# Application of Propolis in Protecting Skeletal and Periodontal Health—A Systematic Review

**DOI:** 10.3390/molecules26113156

**Published:** 2021-05-25

**Authors:** Sophia Ogechi Ekeuku, Kok-Yong Chin

**Affiliations:** Department of Pharmacology, Faculty of Medicine, Universiti Kebangsaan Malaysia, Level 17, Preclinical Building, Jalan Yaacob Latif, Bandar Tun Razak, Cheras 56000, Kuala Lumpur, Malaysia; virgosapphire2088@yahoo.com

**Keywords:** bee wax, honeybees, periodontitis, osteoblast, osteoclast, osteopenia, osteoporosis

## Abstract

Chronic inflammation and oxidative stress are two major mechanisms leading to the imbalance between bone resorption and bone formation rate, and subsequently, bone loss. Thus, functional foods and dietary compounds with antioxidant and anti-inflammatory could protect skeletal health. This review aims to examine the current evidence on the skeletal protective effects of propolis, a resin produced by bees, known to possess antioxidant and anti-inflammatory activities. A literature search was performed using Pubmed, Scopus, and Web of Science to identify studies on the effects of propolis on bone health. The search string used was (i) propolis AND (ii) (bone OR osteoporosis OR osteoblasts OR osteoclasts OR osteocytes). Eighteen studies were included in the current review. The available experimental studies demonstrated that propolis could prevent bone loss due to periodontitis, dental implantitis, and diabetes in animals. Combined with synthetic and natural grafts, it could also promote fracture healing. Propolis protects bone health by inhibiting osteoclastogenesis and promoting osteoblastogenesis, partly through its antioxidant and anti-inflammatory actions. Despite the promising preclinical results, the skeletal protective effects of propolis are yet to be proven in human studies. This research gap should be bridged before nutraceuticals based on propolis with specific health claims can be developed.

## 1. Introduction

The skeletal system consisting of dense connective tissues, mainly bone, is metabolically active and functionally diverse. It undergoes modelling (construction) and remodelling (reconstruction) process in response to stimuli throughout our lifetime [1,2]. Bone loss associated with age is often a result of defective bone remodelling. Bone remodelling refers to the skeletal reparative process, whereby small areas of bone are resorbed by osteoclasts and replaced by osteoblasts to prevent the accumulation of microfractures and preserve mineral homeostasis by releasing calcium and phosphorus into the circulation. This tightly regulated process replaces 10% of the bone annually and ensures that skeletal structural integrity, mass and mechanical strength are preserved [3,4].

Many endogenous and exogenous factors influence the bone remodelling process. Among the factors is oxidative stress resulted from an imbalance between oxidants [reactive oxygen species (ROS) and reactive nitrogen species (RNS)] and antioxidants (enzymatic and non-enzymatic) [5,6]. Oxidative stress favours osteoclast formation (osteoclastogenesis) and bone resorption while hampering osteoblast formation (osteoblastogenesis) and bone formation, leading to bone loss [7,8,9]. These skeletal effects are achieved by activating signalling pathways, such as mitogen-activated protein kinases (MAPKs), including extracellular signal-regulated kinases (ERK1/2), c-Jun-N terminal kinase (JNK), and p38 MAPK [10,11,12]. Given the role of oxidative stress in the development of bone loss, antioxidants present in food could have beneficial skeletal effects. Antioxidants improve the survival and functions of osteoblasts and osteocytes, while suppressing osteoclastogenesis and osteoclast activity [7,10,13,14]. Vitamin C, vitamin E, polyphenols, and other antioxidants have been shown to promote osteoblastogenesis, as well as preventing oxidative stress-induced apoptosis of osteoblasts and osteoclastogenesis [15,16,17,18].

Inflammation is closely associated with oxidative stress. Phagocytes involved in the immune response synthesise ROS to destroy the invading pathogens. These ROS could diffuse to the surrounding tissues causing damage. Both lipopolysaccharide from the bacterial cell wall and interleukin (IL)-4 synthesised by the immune cells could activate nicotinamide adenine dinucleotide phosphate (NADPH) oxidases (NOX) which generate ROS. In reciprocal, ROS can activate nuclear factor kappa B (NF-κB) signalling pathway and nucleotide-binding and oligomerisation domain-like receptor family pyrin domain containing 3 inflammasome [19,20]. As implicated in inflammatory bowel disease, rheumatoid arthritis and systemic lupus erythematosus, chronic inflammation is a strong risk factor for bone loss. The pro-inflammatory cytokines released by immune cells potentiate inflammation and promote bone resorption and impair bone formation, resulting in accelerated bone loss and increased fracture risk [21,22,23].

Propolis (or bee glue) is a natural resin mixture produced by honeybees from plant parts, buds, and exudates. Given its waxy consistency and mechanical properties, propolis is used by the bees in the construction and repair of hives to protect against foreign predators and weather elements [24,25]. Propolis contains over 300 potentially active ingredients, including coumarins, phenolic aldehydes, steroids, amino acids and polyphenols [26]. Its functions as an immune enhancer, antibacterial, anti-inflammatory, anti-tumour and antioxidant agent have been investigated [27,28]. The antioxidant properties of propolis are contributors to its other biological effects, including chemoprevention [29] and anti-inflammation [30]. With regard to its anti-inflammatory effects, propolis could inhibit the synthesis of prostaglandin E2 and the inducible cyclooxygenase-2 expression [25,31,32]. In a model of carrageenin-induced paw oedema, propolis prevented inflammation probably by inhibiting nitric oxide (NO) production [30]. Regarding its antioxidant effects, propolis prevented DNA damage caused by hydrogen peroxide in fibroblasts [33]. It also inhibits protein nitration, peroxidation of low-density lipoprotein and endothelial NOX expression, and increases endothelial nitric oxide synthase expression [34]. Besides, propolis could enhance antioxidant capacity in animals [35] and humans [36], thereby lowering lipid peroxidation, which is linked to an increased risk of cardiovascular diseases [37,38].

Given its antioxidant and anti-inflammatory actions, propolis could protect the skeletal system. The current review aims to summarise the evidence on the skeletal action of propolis. Since bone loss could occur at the systemic level and locally (for instance, at the periodontal region), the effects of propolis on both types of bone loss would be discussed. We hope the review will facilitate the practical application of propolis in enhancing bone and periodontal health.

## 2. Results

### 2.1. Selection of Articles

The literature search yielded 140 articles (69 from PubMed, 48 from Scopus, and 23 from Web of Science (WoS). After removing 46 duplicates, 94 unique articles were screened. A total of 76 articles were eliminated for not meeting the selection criteria (1 commentary, 6 articles not written in English, 12 review articles, and 57 articles with topics not relevant to the current review) (Appendix A). Finally, 18 articles fulfilling all criteria mentioned were included in the current review. 

### 2.2. Study Characteristics 

The selected studies were published between 2008 and 2020. For in vitro osteoclastogenesis studies, mouse marrow cells, human peripheral blood mononuclear cells, and RAW264.7 cell line were the cell models used. Mouse marrow and RAW264.7 cell were stimulated with receptor activator of NF-ĸB (RANK) ligand (RANKL) and cultured with 1–10 µL of propolis [39], while human peripheral blood mononuclear cells were stimulated with macrophage colony-stimulating factor (M-CSF) and RANKL and cultured with 0.025–10 mg/mL of propolis [40]. For osteoblastogenesis studies, the cell models used were MC3T3-E1 and MG-63 cell lines. MC3T3-E1 pre-osteoblast cell line was unstimulated and cultured on 60% propolis-loaded plates [41], while MG-63 human osteoblast cell line was stimulated by ascorbic acid and β-glycerophosphate, and cultured with 0.017–0.034 mg/mL propolis [42]. The treatment period for osteoclast differentiation was 1–7 days, and for osteoblast differentiation was 1–14 days.

The animal model used to investigate the skeletal effects of propolis could be divided into systemic bone loss models, fracture healing models and periodontal models. Al-Hariri et al. [43] examined the effects of propolis (300 and 600 mg/kg propolis) in Wistar rats with streptozotocin (STZ)-induced diabetes. The bone healing properties of propolis were explored using critical non-union bone defects in Wistar rats (dose of propolis: 0.1 mL of 250 mg/mL propolis percutaneously with chitosan or bone graft) [44], femur fracture in Sprague Dawley rats (implanted with 60% propolis loaded implants and treated with 200 mg/kg of propolis) [45], distraction osteogenesis in New Zealand white rabbits (dose of propolis: 100 and 200 mg/kg) [46]. For the periodontal models, periodontitis-induced bone loss in Wistar rats and C57BL/6 mice (dose of propolis: 100–200 mg/kg) [47,48,49], orthodontic tooth movement (OTM) in Wistar rats and guinea pigs (dose of propolis: 2–5% and 100 µL of propolis) [50,51,52]; delayed tooth replantation in Wistar rats (dose of propolis: 200 mL of 15% propolis solution) [53], rapid maxillary expansion (RME) in Wistar rats (dose of propolis: 100 mg/kg or propolis) [54]; grade II furcation-induced bone loss in mongrel dogs (dose of propolis: 400 mg in graft) [55] were studied. In animal studies, densitometry method was used to measure bone mass, micro-computed tomography and bone histomorphometry were used to determine bone microstructure, and circulating bone markers were used to estimate the bone remodelling. No human studies were reported on this topic.

Overall, 2 studies showed that propolis suppressed osteoclastogenesis [39,40], while 2 studies showed that propolis promoted osteoblastogenesis [41,42]. In the animal experiments, propolis inhibited bone loss due to periodontitis in 2 studies [48,49], bone loss due to periodontitis/STZ-induced diabetes in 1 study [56], bone loss due to OTM in 3 studies [50,51,52] and bone loss due to dental implantitis in 1 study [41]. Propolis prevented systemic bone loss due to STZ-induced diabetes in 1 study [43]. Propolis also demonstrated beneficial effects in distraction osteogenesis in 1 study [46], RME in 1 study [54], delayed tooth replantation in 1 study [53]. Two studies showed that propolis promoted fracture healing of the femur [45] and non-union defect of the radial bone [44]. Three studies demonstrated no beneficial effects of propolis on inflammatory resorption, replacement resorption and extent of fusion between alveolar bone and cementum in delayed tooth replantation [53]; new bone formation in grade II furcation defect [55]; and alveolar bone loss [47]. Table 1 summarises the effects of propolis on the bone health system.

## 3. Discussion

Osteoblasts and osteoclasts are two critical cell types in governing the bone remodelling processes. The current bone-modulating drugs primarily target these cells to achieve their therapeutic effects [57]. Antiresorptive drugs suppress the formation, function and survival of osteoclasts, thus slowing bone resorption, whereas anabolic drugs promote bone formation activities and increase bone mass [58]. Propolis is reported to affect both osteoclasts and osteoblasts. Pileggi et al. [39] showed that propolis reduced the formation of osteoclast-like cells (multinucleated, tartrate-resistant acid phosphatase (TRAP) positive cells) from RAW 264.7 murine macrophages and mouse bone marrow cells. Similarly, propolis also reduced TRAP-positive cells generated from human peripheral blood mononuclear cells (hPBMCs) [40]. In hPBMCs, the reduced osteoclastogenesis was associated with lower expression of osteoclast specific genes, such as receptor activator of nuclear factor kappa B (RANK), nuclear factor of activated T cells 2 (NFAT2), cathepsin K, chloride channel 7 (CLCN7), and calcitonin receptor (CTR) [40]. Other studies demonstrated that propolis promoted osteoblast differentiation and activity. MC3T3-E1 murine osteoblast-like cells cultured on propolis loaded titanium oxide nanotubes showed increased expression of alkaline phosphatase (ALP) and osteoblast differentiation [41]. Propolis increased mineralisation and ALP activity in human osteoblast-like MG-63 cell line [42]. Besides, propolis promoted osteoblast differentiation by increasing expression of runt-related transcription factor 2 (RUNX2), osterix (OSX), osteocalcin, and type 1 collagen alpha [42]. The dual effects of propolis on bone formation and resorption highlight its potential in preventing osteoporosis.

Animal studies showed that propolis could help to manage systemic bone loss. Diabetes mellitus is a risk factor for bone fracture [59]. STZ is a chemotherapeutic agent that damage beta-cells in the pancreas specifically and induce diabetes [60]. In a study by Al-Hariri et al. [43], STZ-induced diabetic rats treated with propolis (300 and 600 mg/kg for 6 weeks) showed an increased ratio of femur weight to body weight and bone calcium and phosphorus level than the diabetic control, suggesting increased bone mass and mineral content. Propolis also decreased parathyroid hormone (PTH) level and normalised calcitonin level in the rats. Therefore, propolis reduced the effects of insulin deficiency and PTH on bone [43].

Fracture is the most important and devastating sequela of osteoporosis [61]. Delayed fracture healing could incur significant medical cost and morbidity to the patients [62]. A study by Guney et al. [45] showed that propolis treatment (200 mg/kg/day for 3 weeks and 6 weeks) increased bone mineral density (BMD), radiological and histological scores in rats with experimental fracture and retrograde fixation. Propolis also reduced the levels of endogenous antioxidants such as superoxide dismutase (SOD), myeloperoxidase and glutathione, suggesting that propolis could scavenge free radicals without invoking adaptive endogenous antioxidant response [45]. Meimandi-Parizi et al. [44] observed increased formation of new bone tissue, woven bone, and cartilage tissue in rats with critical non-union bone defects in the radius of male Wistar rats treated with demineralised bone matrix and propolis (0.1 mL of 250 mg/mL of propolis; days 0 and 3). Propolis also increased maximum load, maximum stress, yield load and decreased ultimate strain and yield strain in radius/ulna complex of rats with critical non-union bone defects [44].

The skeletal protective effects of propolis can be extended to periodontal tissues. Alveolar bone loss due to excessive bone resorption could impair tooth support, leading to tooth loss [63]. It is a hallmark of periodontitis and a major clinical challenge in managing periodontal diseases [64]. In male Wistar rats with ligature-induced periodontitis/STZ-induced diabetes, propolis treatment reduced alveolar bone loss marked by a decreased distance between cementoenamel junction and alveolar bone crest. However, plasma IL-1β, tumour necrosis factor-alpha (TNF-α), and metalloproteinase 8 (MMP-8) levels were not altered significantly by the induction of periodontitis or propolis treatment [56]. Since the changes in circulating biochemical markers might be transient, the lack of changes could be due to the sampling time point. In male Wistar rats with alveolar bone loss caused by ligature-induced periodontitis, propolis treatment (100 and 200 mg/kg for 11 days) reduced osteoclast number in alveolar bone and reduced alveolar bone loss [48]. Yuanita et al. [49] also reported that propolis (10 μL) reduced osteoclast number and increased osteoprotegerin (OPG) expression in periapical area of alveolar bone in rats with *Enterococcus faecalis*-induced chronic apical periodontitis. A furcation defect occurs due to excessive bone resorption at the bi- or trifurcation area of a multi-rooted tooth in the presence of periodontal disease [65]. Adult male mongrel dogs with surgically created grade II furcation defects showed increased trabecular bone, bone height, and surface area when the furcation defect was filled with propolis (400 mg for 1 and 3 months). However, propolis did not increase the amount of newly formed alveolar bone than the positive control [55]. Similarly, propolis treatment (200 mg/kg of propolis for 5 weeks) exerted no beneficial effect on alveolar bone loss in mice administered with *Porphyromonas gingivalis* orally. The lack of significance might be due to the failure to induce prominent periodontitis via this method compared to ligation [47].

OTM is based on synchronised tissue resorption and formation in the surrounding bone and periodontal ligament. Tooth loading in OTM induces local hypoxia and fluid flow, triggering an aseptic inflammatory cascade. This event results in osteoclast bone resorption in compression areas and osteoblast matrix deposition in stress areas [66]. In two studies by Kresnoadi et al., male guinea pigs with OTM treated with propolis (2% propolis in polyethylene glycol; 0.1 mL propolis extract) for 3 and 7 days showed increased bone formation at the alveolar bone, as evidenced by increased osteoblast number and protein expression of osteocalcin. Propolis also reduced bone resorption at the alveolar bone, as evidenced by decreased osteoclast number [50,51]. In another study by Wiwekowati et al. [52], propolis treatment (5% propolis gel for 17 days) showed increased osteoblast number in the alveolar bone of rats with OTM. Propolis also reduced oxidative damage, as evidenced by decreased lipid peroxidation product, MDA, in the blood [52]. RME is a common orthodontic procedure to increase the transverse width of the maxillary basal bone but studies have reported root resorption following the procedure [54,67]. In a study by Altan et al. [54], rats with RME and treated with propolis (100 mg/kg/day for 12 days) showed increased bone formation marked by a higher osteoblast number and new the maxillary bone. Concurrently, osteoclast number was also elevated with treatment, which the authors associated with the coupling effects of bone remodelling [54]. They also reported increased capillary number and inflammatory cell infiltration in the maxillary bone with treatment [54].

Distraction osteogenesis is a new procedure for restoring atrophic alveolar bone before implant placement [68]. Bereket et al. [46] reported that propolis treatment (200 mg/kg/day for 32 days) increased area of matured bone (evidenced by decreased volume of new bone area), BMD and bone mineral content in distraction gap of mandible bone of rats with distraction osteogenesis. However, propolis treatment decreased new bone formation (evidenced by a decreased number of osteoblast cell lining the bone surface) and showed no change in volume of connective tissue and number of capillaries in the distraction gap of mandible bone [46].

Bone resorption at the area of dental implantitis is a major change in prosthetic rehabilitation [69,70]. Rats implanted with propolis-loaded titanium oxide nanotubes showed increased new bone formation and bone mineral density around implants in mandibular bone compared to titanium oxide nanotubes that were not loaded with propolis [41]. Propolis increased expression of collagen fibres and osteogenic differentiation marked by increased expression of bone morphogenic protein (BMP)-2 and 7. Propolis also reduced inflammation around the implant surface, marked by reduced IL-1β and TNF-α expression [41]. These results showed that propolis could improve bone formation and reduce bone resorption, oxidative stress, and inflammation, thereby improving bone remodelling that is very important in the implant osteointegration process. 

Delayed replantation of permanent tooth that has been completely displaced could result in external root resorption [53]. In a study by Gulinelli et al. [53], propolis (20 mL of 15% propolis in propylene glycol solution for 10 min) had no effect on the inflammatory resorption, replacement resorption and extent of fusion between alveolar bone and cementum in rats with delayed tooth replantation.

Our literature search did not find any human studies on the skeletal effects of propolis. A search through https://clinicaltrials.gov/ (accessed on 2 April 2021) revealed 64 registered trials on the effects of propolis on various conditions (Appendix A). Of the 22 registered trials on dental diseases, only two studies measured the effects of propolis on the pocket depth of alveolar bone (identifier: NCT02794506) and periapical bone density (identifier: NCT03533231). Both trials have been completed, but no results have been made available. 

### 3.1. Possible Molecular Mechanisms of Propolis in Preserving Skeletal Health

Many studies have shown that antioxidants help to promote osteoblast differentiation, mineralisation and reduce osteoclast activity by directly or indirectly counteracting the effects of oxidants [13,14,71]. Propolis has been shown to possess antioxidant activity [72], and the majority of studies showing a reduction in oxidative stress markers with propolis treatment [73]. Guney et al. [45] reported a reduction in femoral SOD and glutathione levels in rats treated with propolis. The authors suggested that the administration of an exogenous antioxidant could reduce endogenous antioxidant enzyme expression because the components in propolis could scavenge free radicals [45]. Malondialdehyde (MDA) is frequently used as an indicator of oxidative lipid damage. Wiwekowati et al. [52] reported that propolis reduced MDA levels in rats with OTM. Propolis could effectively eliminate free radicals due to the polyphenol content [74]. Flavonoids, one of the polyphenols in propolis, are potent antioxidants capable of scavenging free radicals, thereby shielding the cell membrane from lipid peroxidation [75]. A positive relationship has been established between the flavonoid content and propolis inhibition of MDA [76]. Caffeic acid phenethyl ester (CAPE) has also been linked to the antioxidant properties of propolis [18]. Propolis extracts containing CAPE were more efficient at inhibiting xanthine oxidase and lipoperoxidase activity than propolis extracts lacking CAPE [77]. The presence of caffeic acid and phenyl caffeate is linked to the high antioxidant potential of propolis [78]. 

M-CSF and RANKL are two essential cytokines regulating osteoclast differentiation. M-CSF ensures the survival and proliferation of osteoclast precursor cells. It also increases the RANK expression in osteoclast precursor cells, ensuring a more efficient response to the RANKL-RANK signalling pathways [79,80,81,82]. Wimolsantirungsri et al. [40] reported decreased RANKL and M-CSF-induced RANK expression in osteoclast precursor cells following propolis treatment, suggesting the inhibition of RANKL-RANK signalling pathway, which would eventually lead to reduced osteoclast differentiation. RANKL activates several transcription factors, including NFAT2, a master regulator of osteoclast differentiation that controls the expression of several osteoclast-specific genes, such as TRAP, cathepsin K, and CTR [83,84,85,86]. Two studies demonstrated that propolis inhibited osteoclast formation from human peripheral blood mononuclear cells [40] and RAW 264.7 cells [39] by inhibiting the expression of NFAT2, cathepsin K and CTR.

Inflammatory cytokines cause bone loss by promoting osteoclast differentiation and maturation directly. These cytokines work together to attract, differentiate, and activate osteoclasts through the NF-κB signalling pathway [87,88,89]. Propolis-incorporated bone implants were shown to downregulate the expression of IL-1β, and TNF-α at the surrounding tissue [41], thereby preventing osteoclast formation and bone resorption that would loosen the implants. Another study on ligature-induced periodontitis failed to reduce circulating inflammatory cytokines by propolis, probably because the rats also presented diabetes and a higher degree of inflammation [56]. Propolis suppressed LPS-induced expression of IL-1β, IL-6, and IL-8 in human periodontal ligament cells [90] and IL-6 in RAW264.7 macrophages [91].

Osteoblasts derived from mesenchymal stem cells in the bone marrow are responsible for the synthesis, secretion, and mineralisation of bone matrix. They also secrete OPG, a RANKL decoy receptor that prevents the binding of RANKL to RANK, thereby halting RANKL signalling and osteoclastogenesis [92]. Propolis was shown to stimulate the proliferation, differentiation and maturation of osteoblasts [42]. Among the many osteoblast markers, OPG expression was also upregulated by propolis, indicating that it could affect RANKL/OPG dynamic or crosstalk between osteoblasts and osteoclasts, thereby suppressing osteoclastogenesis. 

BMPs are growth factors belonging to the transforming growth factor-superfamily. Their diverse functions range from regulating bone induction, preservation, and reconstruction to determining non-osteogenic embryological developmental pathways and maintaining adult tissue homeostasis [93]. BMP-2 and 7 are involved in bone development and regeneration during osteoblast differentiation [94]. The role of BMP signalling in polyphenol-mediated bone anabolism has been extensively studied, and several studies have shown that increasing BMP-2 promoter activity and BMP-2 expression increases new bone formation [95,96]. Propolis loaded implants increased the expression of BMP-2 and 7 at the surrounding tissue. This event occurs in parallel with a reduction in inflammatory cytokine expression, increased new bone development around the implant, and enhanced adhesion with the mandibular and implant [41]. The possible molecular mechanisms of propolis are summarised in Figure 1.

Based on current evidence and the possible molecular mechanism, propolis has the potential to protect the skeletal system and improve bone remodelling by reducing the expression of inflammatory cytokines responsible for osteoclast differentiation and osteoblast apoptosis, inhibiting RANKL and M-CSF signalling pathway responsible for osteoclast differentiation and maturation and increasing osteoblast proliferation, differentiation, and mineralisation through its increased expression of osteoblast markers. These properties could be useful in the treatment of several medical conditions which promote bone loss or fractures. So far, propolis is already being considered in periodontal healthcare as there are existing clinical trials on the effect of propolis on periodontal health. However, there are no results available.

### 3.2. Bioavailability and Safety Concerns of Propolis

Propolis is made up of lipids, waxes, and resins in a complex matrix with a high molecular weight, contributing to its low absorption and bioavailability [97]. The type of polyphenols present and their interactions dictate the synergistic effects and influence the bioavailability of propolis [97]. Digestive instability, poor transcellular efflux in intestinal cells, as well as rapid metabolism and excretion are all thought to play a role in polyphenol bioavailability [75]. Dietary polyphenols cannot be absorbed because they exist as esters, polymers, or glycosylated forms, and must be hydrolysed by intestinal enzymes or colonic microflora before absorption [97]. Poorly absorbed polyphenolic compounds are converted to smaller phenolic acids with improved bioavailability in the intestinal system, owing to the colonic microbiota enzyme activity [98]. Inter-individuality in absorption and metabolism is important as individual microbiota differ. Despite the low absorption percentages of bio-accessible phenolic compounds in propolis, Yesiltas et al. [99] reported that the recovered amounts detected in plasma were still high compared to other food materials like fruits and vegetables. 

Propolis and its constituents are generally well-tolerated and non-toxic unless given in huge doses according to clinical studies in mice and humans [74,100,101]. According to Dobrowolski et al. [102], the median lethal dose (LD50) for various propolis sources ranged from 2 to 7.3 g/kg in mice, implying a healthy dosage of 1.4 to 70 mg/day for humans based on a safety factor of 1000. The LD_50_ of propolis extract given to conscious mice was more than 7.34 g/kg, indicating that the product is generally safe [103,104]. However, it should be noted that propolis toxicity and adverse events were rarely monitored in human trials. Hypersensitivity is a more common side effect of propolis, especially when applied topically, resulting in allergic reactions, swelling, dermatitis, and urticaria [105]. Hsu et al. [105] reported a patient presenting severe swelling of the throat and anaphylactic shock upon topical application of propolis. Severe side effects like laryngeal oedema and anaphylactic shock have been reported infrequently [106] and are rarely attributed to propolis. Propolis sensitivities have been reported in 1.2–6.6% of people with dermatitis [107]. Therefore, consumers should seek medical advice before taking propolis supplements or applying propolis products, despite its positive safety profile.

The studies examined revealed several common limitations. Half of the studies reviewed did not use a positive control to compare the effect of propolis on bone. As a result, it is impossible to compare the therapeutic effects of propolis and currently available anti-resorptive therapy. Due to the lack of a human clinical trial, the effects of propolis in humans cannot be confirmed. Future research can improve these aspects. There are several limitations pertaining to the current review. Non-English or non-indexed articles may be overlooked because we only consider articles written in English and published in Scopus, Pubmed, and WoS.

## 4. Materials and Methods

### 4.1. Literature Review

The systematic literature search was performed in February 2021 using three electronic databases: Pubmed, Scopus, and WoS to identify studies on the effects of propolis on bone health. The search string used was (i) propolis AND (ii) (bone OR osteoporosis OR osteoblasts OR osteoclasts OR osteocytes). 

### 4.2. Selection of Research Articles

Only articles written in English language were considered. Studies with these characteristics were included: (i) original research articles with the primary objective of determining the effects of propolis on bone health; (ii) studies using cellular or animal models, or human (iii) studies administering propolis as a single treatment agent. Articles were excluded if they were (i) letter, commentary, editorial, perspectives, review or conference abstract; (ii) written in other languages; (iii) using isolated compounds from propolis. Mendeley software (Elsevier, London, UK) was used to organise the search results. Duplicates were identified using Mendeley and manually.

### 4.3. Data Extraction 

Two authors screened the same databases using the search string mentioned. After collating all studies, the authors screened the titles and abstracts of the articles for relevant studies. After removing irrelevant studies, the authors examined the full text of remaining studies and matched them with the inclusion and exclusion criteria. The two authors compared the list of included studies and resolved any discrepancy by discussion. The data extracted included researchers (year of publication), study characteristics (model, dose of propolis and negative/positive control), and major findings. The data were tabulated in a standardised evidence table (Table 1). The Preferred Reporting Items for Systematic Reviews and Meta-Analyses (PRISMA) guidelines and checklist were used to conduct this systematic review [108]. The flowchart in Figure 2 depicts how the results were sorted.

## 5. Conclusions

Propolis supplementation improves bone remodelling and protects skeletal health by increasing osteoblastogenesis and reducing osteoclastogenesis. These effects are exerted through its anti-inflammatory and antioxidant properties. However, due to a lack of human studies, it is unclear whether these findings can be applied to humans. A well-designed randomised control trial should be conducted to confirm the effectiveness of propolis supplementation on bone health in humans.

## Figures and Tables

**Figure 1 molecules-26-03156-f001:**
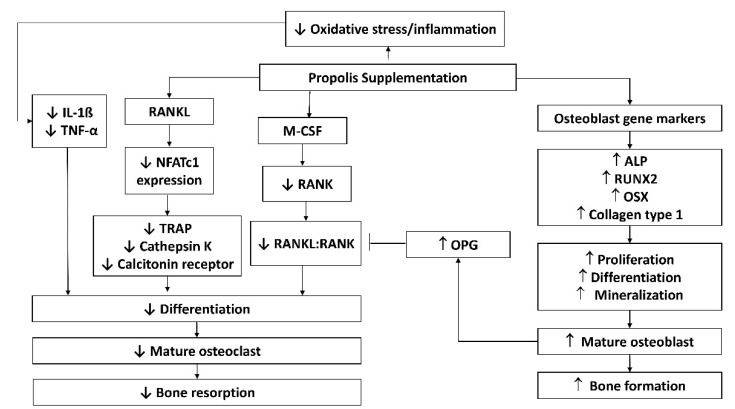
Possible molecular mechanisms of propolis in protecting against bone loss. Abbreviations: ALP, alkaline phosphatase; IL-1β, interleukin-1 beta; M-CSF, macrophage colony-stimulating factor; NFATc1, nuclear factor of activated T-cells; OPG, osteoprotegerin; OSX, osterix; RANK, receptor activator of nuclear factor-kappa B; RANKL, RANK ligand; TNF-α, tumour necrosis factor-alpha; TRAP, tartrate-resistant acid phosphatase.

**Figure 2 molecules-26-03156-f002:**
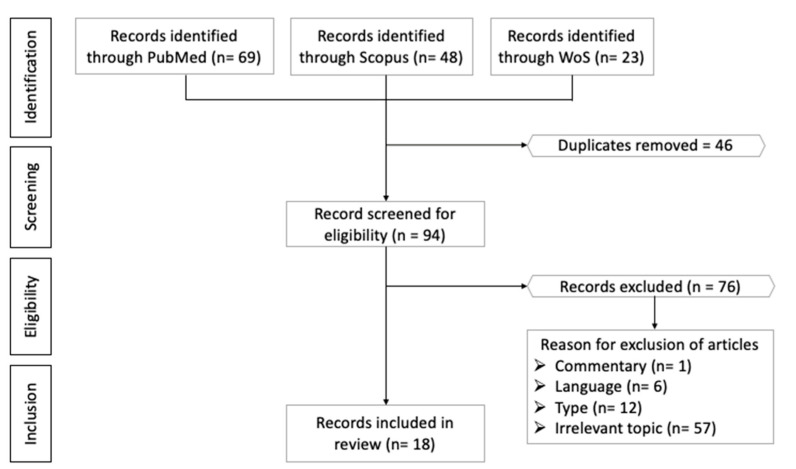
Process of article selection.

**Table 1 molecules-26-03156-t001:** Proof of propolis for protection of bone health.

Researcher	Study Design	Findings
Cell Culture Studies		
Pileggi et al. [39]	Cell: RAW 264.7 and mouse marrow cells Induction: RANKL-induced osteoclastogenesis Treatment: 1 and 10 μL propolis for 7 days Control: Negative: no treatment Positive: no	↑ mononuclear TRAP+ cells vs. negative control ↓ multinuclear and giant TRAP+ cells vs. negative control ↓ actin rings formation vs. negative control
Wimolsantirungsri et al. [40]	Cell: Human peripheral blood mononuclear cells Induction: RANKL and M-CSF-induced osteoclastogenesis Treatment: 0.025–10 mg/mL propolis for 24 h or 7 days Control: Negative: no treatment Positive: no	↓ TRAP-positive cells with the range 0.1–10 mg/mL vs. negative control ↓ expression of osteoclast-specific genes (NFAT2, CTSK, RANK, CLCN7, and CTR) in dose dependent manner in 0.1 and 0.5 mg/mL vs. negative control
Somsanith et al. [41]	Cell: MC-3T3-E1 pre-osteoblasts Induction: Treatment: 60% crude propolis-loaded TNT-Ti (PL-TNT-Ti) plates for 1, 5 and 14 days Control: Negative: CP-Ti Positive: TNT-Ti	↑ cell proliferation and ALP activity vs. negative control
Lim et al. [42]	Cell: Human osteoblast-like cell line MG-63 Induction: NA Treatment: 17 and 34 μg/mL propolis extract for 24 h Control: Negative: no treatment Positive: 10 ng/mL rhBMP2	↑ mineralisation and ALP activity in 34 µg/mL propolis-treated group vs. negative control ↑ RUNX2 expression on day 2 and 8 in 34 µg/mL propolis-treated group and day 8 in 17 µg/mL propolis group compared to negative control ↑ OSX expression on day 4 in 17 and 34 µg/mL propolis treated group vs. negative control; ↓ OSX expression on day 8 in 34 µg/mL propolis-treated group vs. negative control ↓ type 1 collagen alpha expression on day 2 in 34 µg/mL propolis-treated group, day 2 and 8 in 17 µg/mL propolis group and ↑ on day 1 in 17 µg/mL vs. negative control ↑ osteocalcin expression on day 2, 4 and 8 in 17 µg/mL propolis group vs. negative control
Animal Studies		
Toker et al. [48]	Animals: Male Wistar rats (300–330 g) Disease Model: Ligature-induced periodontitis Treatment: 100 and 200 mg/kg of propolis (oral gavage) for 11 days Control: Negative: no treatment Positive: no	↓ alveolar bone loss vs. negative control ↓ osteoclast number in alveolar bone vs. negative control
Gulinelli et al. [53]	Animals: Male Wistar rats (250–300 g) Disease Model: Delayed tooth replantation Treatment: Extracted teeth was immersed in 20 mL of 15% propolis in propylene glycol solution for 10 min before replantation Control: Negative: 20 mL physiologic saline Positive: 20 mL of 2% acidulated phosphate sodium fluoride	⟷ inflammatory resorption in alveolar bone vs. positive and negative control ⟷ replacement resorption in alveolar bone vs. positive and negative control ⟷ extent of fusion between alveolar bone and cementum vs. positive and negative control
Al-Hariri et al. [43]	Animals: Adult male albino rats (150–300 g) Disease Model: STZ-induced diabetes Treatment: 300 and 600 mg/kg of propolis (oral gavage) for 6 weeks Control: Negative: no treatment Positive: Insulin injection (5 IU/kg/day)	↓ calcitonin and PTH in plasma vs. negative control ⟷ ratio of femur ash to femur weight and magnesium in femur ash vs. negative control and positive control ↑ calcium and phosphorus in femur ash vs. negative and positive control ↑ femur weight to body weight ratio vs. negative control
Guney et al. [45]	Animals: Male Sprague Dawley rats (280–480 g) Disease Model: Femur fracture and retrograde fixation Treatment: 200 mg/kg/day of propolis (oral gavage) for 3 weeks and 6 weeks Control: Negative: no treatment Positive: no	↑ bone mineral density vs. negative control ↑ radiological and histological scores in femur vs. negative control ↓ plasma SOD at week 3 vs. negative control ⟷ plasma SOD at week 6 vs. negative control ↓ SOD in bone tissue at weeks 3 and 6 vs. negative control ↓ total GSH and MPO levels in plasma and bone tissue at weeks 3 and 6 vs. negative control
Altan et al. [54]	Animals: Male Wistar albino rats 200 g (±10 g) 12 weeks old Model: RMET reatment: 100 mg/kg/day of propolis (oral gavage) for 12 days. Control: Negative: no treatment Positive: no	↑ osteoclast, osteoblast and capillary numbers in maxillary bone vs. negative control ↑ new bone formation and inflammatory cell infiltration in maxillary bone vs. negative control
Bereket et al. [46]	Animals: Male New Zealand white rabbits (2.5–3.0 kg). Model: Distraction osteogenesis Treatment: 100 (P100) and 200 (P200) mg/kg/day of propolis (oral gavage) for 32 days. Control: Negative: no treatment Positive: no	↓ new bone formation in distraction gap of mandible bone vs. control group. ↑ area of matured bone in distraction gap of mandible bone in P200 vs. P100 and control group. ⟷ volume of connective tissue (Vct), number of capillaries (Nc) in distraction gap of mandible bone vs. control group. ↓volume of new bone area (Vn) in distraction gap of mandible bone of P200 vs. P100 and control group ↑ BMC for P200 at week 1 and 4 vs. P100 and control group ↑ BMD for P200 at week 1 and 4 vs. P100 and control group
Aral et al. [56]	Animals: Male Wistar albino rats (300–350 g) Disease Model: Ligature-induced periodontitis/STZ-induced diabetes Treatment: 100 mg/kg/day of propolis (oral gavage) for 21 days. Control: Negative: no treatment Positive: no	↓ alveolar bone loss vs. negative control ⟷ plasma IL-1β, TNF-α, and MMP-8 levels vs. negative control ↓ linear distance from cementoenamel junction to the alveolar bone crest vs. negative control.
Nakajima et al. [47]	Animals: Male C57BL/6 mice (8 weeks old) Disease Model: *Porphyromonas gingivalis*-induced periondotitis Treatment: 200 mg/kg propolis (oral gavage) for 5 weeks. Control: Negative: no treatment Positive: no	⟷ alveolar bone loss vs. negative control
Somsanith et al. [41]	Animals: Male Sprague-Dawley rats Model: Dental implantation Treatment: Crude propolis extract (purity 60%)-loaded TNT-Ti implants (PL-TNT-Ti) for 1 and 4 weeks Control: Negative: TNT-Ti implants Positive: no	↑ new bone formation around implants in mandibular bone vs. negative control at 4 weeks ↑ bone mineral density and the volume of newly formed bone around implants in mandibular bone vs. negative control at 1, 2, 3, and 4 weeks ↑ expression of well-formed collagenous bone trabeculae, muscle fibres and cytoplasm around implants in mandibular bone vs. negative control ↑ formation of new bone with concentration of macrophages and nuclei around implant surface in mandibular bone vs. negative control ↓ expression of inflammatory cytokines such as IL-1β, and TNF-α around the surface of the implant in mandibular bone vs. negative control. ↑ expression of bone formation molecules BMP-2 and 7 around the surface of the implant in mandibular bone vs. negative control
Yuanita et al. [49]	Animals: Male Wistar rats (130–150 g) Disease Model: *Enterococcus faecalis-*induced chronic apical periodontitis Treatment: 10 μL of 12% propolis aqua destilata (pure water) Control: Negative: no treatment Positive: no	↓ osteoclast number and ↑ OPG expression in periapical of alveolar bone vs. negative control
Zohery et al. [55]	Animals: Male Mongrel dogs (18–24 months old, 18–24 kg) Disease Model: Surgically created grade II furcation defects Treatment: 400 mg propolis graft for 1 and 3 months Control: Negative: no Positive: nanohydroxyapatite graft	⟷ newly formed bone in alveolar bone after 1 month vs. positive control ↑ trabecular bone in inter-radicular defect after 3 months vs. positive control ↑ bone height and surface area of inter-radicular bone vs. positive control
Meimandi-Parizi et al. [44]	Animals: Male Wistar rats (8 weeks old, 200–250 g) Disease Model: critical non-union bone defect Treatment: 0.1 mL of 250 mg/mL propolis extract (injected percutaneously into defect site) on day of operation and day 3 post operation (chitosan-propolis and DBM-propolis) Control: Negative: no treatment Positive: Chitosan and DBM scaffolds	↑ formation of fresh bone tissue, woven bone and cartilage tissue in radius and ulna complexes of DBM-propolis group vs. negative control, positive control and chitosan-propolis group ↑ maximum load, maximum stress and yield load and ↓ ultimate strain and yield strain in radius and ulna complexes of DBM-propolis group vs. negative control, positive control and chitosan-propolis group
Wiwekowati et al. [52]	Animals: Male Wistar rats (200–250 g) Disease Model: Orthodontic tooth movement Treatment: 5% propolis in carboxymethyl cellulose/nipagin/glyceryl/triethanolamine gel mixture for 17 days Control: Negative: no treatmentPositive: no	↑ osteoblast number in alveolar bone vs. negative control ↓ serum MDA level vs. negative control
Kresnoadi et al. [50]	Animals: Male guinea pigs (3–3.5 months, 300–350 g) Disease Model: Orthodontic tooth movement Treatment: 100 µL (0.1 cc) of propolis extract (filled into alveolar bone socket) for 3 and 7 days Control: Negative: polyethylene glycol Positive: bovine bone graft	↑ osteoblast number in alveolar bone vs. negative control
Kresnoadi et al. [51]	Animals: Male guinea pigs (3–3.5 months, 300–350 g) Disease Model: Orthodontic tooth movement Treatment: 2% (0.5 g) propolis in polyethylene glycol (filled into alveolar bone socket) for 3 and 7 days Control: Negative: polyethylene glycol Positive: bovine bone graft	↑ osteoblast number, osteocalcin expression and ↓ osteoclast number in alveolar bone vs. negative control

Abbreviations: ↑, increase or upregulate; ↓, decrease or down-regulate; ⟷, no change; ALP, alkaline phosphatase; BMC, bone mineral content; BMD, bone mineral density; BMP 2 and 7, bone morphogenic protein 2 and 7; CP-Ti, commercially pure titanium; CLCN7, chloride channel 7; CTR, calcitonin receptor; CTSK, cathepsin K; DBM, demineralised bone matrix; GSH, glutathione; IL-1β, interleukin 1 beta; M-CSF, macrophage colony stimulating factor; MDA, malondialdehyde; MMP-8, metalloproteinase-8; MPO, myeloperoxidase; Nc, number of capillaries; NFAT2, Nuclear factor of activated T cells 2; OPG, osteoprotegerin; OSX, osterix; PL-TNT-Ti, propolis loaded titanium oxide nanotubes on titanium plates/implants; PTH, parathyroid hormone; RANK, receptor activator of nuclear factor kappa B; RANKL, receptor activator of nuclear factor kappa B ligand; rhBMP2, recombinant human bone morphogenetic protein; RME, rapid maxillary expansion; RUNX2, runt-related transcription factor 2; SOD, superoxide dismutase; STZ, streptozotocin; TNF-*α*, tumour necrosis factor alpha; TNT-Ti, titanium oxide nanotube on titanium plates/implants; TRAP, tartrate-resistant acid phosphatase; TRAP+, tartrate-resistant acid phosphatase positive; Vct, volume of connective tissue; Vn, volume of new bone area.

## Data Availability

The study did not report any data.

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
