# Peer review of "Application of Propolis in Protecting Skeletal and Periodontal Health—A Systematic Review"

_molecules, 2021, doi:10.3390/molecules26113156_

Round 1

Reviewer 1 Report

This review concerning the application of propolis in protecting skeletal and periodontal health, is well referenced and argued. The process used to develop this review is detailed. The mechanisms and phenomena involved in the activity of propolis are also well described. Consequently, I give a favorable opinion to the publication of this review in its current form.

Author Response

Thank you for the kind comments. No reply is required.

Reviewer 2 Report

Dear authors,

I consider your work an important contribution to the study of the medical use of propolis.
However, I find some weakness in the use of terminology, especially in the field of dentistry.
I evaluated the work based on the PRISMA checklist and I just would recommend the record of the systematic review and the  risk of bias assessment.

Abstract

Line 18, dental implantitis

Keywords: I propose bone loss (osteopenia and osteoporosis)  instead of  skeleton

Introduction

Line 27 - Skeletal system consisting of bones whose main tissue is bone.

Line 157-158 -  , propolis inhibited .....OTM in 3 studies [51-53], dental implants in 1 study [42].  I could not really understand what you mean with this sentence !

Discussion

This sentences seems incompletes, please better explain these ideas:

Line 185 - Antiresorptive drugs reduce bone resorption by osteoclasts, ....while anabolic drugs enhance bone formation .... [58]

Line 208 - Propolis also decreased parathyroid and calcitonin level in the rats, indicating that it decreased high bone remodelling [44]. 

Line 229-231 - However, plasma IL-1β, tumour necrosis factor-alpha (TNF-α), and metalloproteinase 8 (MMP-8) levels were not altered by propolis treatment [57].

Line 235 -  in periapical area of alveolar bone

Line 217 - free radicals

Line 224 - periodontal tissues

Line 241 - newly formed alveolar bone than the positive control [56]

Could you please try to justify why it happens?

Line 242 - Similarly, propolis treatment (200 mg/kg of propolis for 5 weeks) had no beneficial effect on alveolar bone loss in mice with Porphyromonas gingivalis-induced periodontitis [48]

Line 281 - improving bone remodelling that is so important in the implant osteointegration process. 

Author Response

Dear reviewer,

Thank you for reviewing our manuscript. We appreciate the constructive comments provided and have responded to each of them. Revisions in the manuscript have been tracked using the “track changes” function. 

Reviewer 3 Report

The review by S.O Eleuku and K.Y. Chin concerns a systematic analysis on the effect of propolis in bone and dental protections. It covers the literature on this topic between 2008 and 2020.

In the manuscript the authors describe very well how they selected the reviewed material and also they clearly explain the reasons that led to their decision not to consider 76 out of 94, i.e only 18.

Table 1 is very useful for the reader representing an excellent way to summarize the most important biomedical results contained in these 18 scientific publications.

In the section “Discussion” these results are contextualized and discussed within a wider scientific area, citing about 50 other scientific paper not strictly related to the subject in question but clearly necessary to understand it.

The effort made by the authors to summarize them in Figure 2 is also noteworthy.

Finally, I greatly appreciated the intellectual honesty of the authors in not over-emphasizing the healing abilities of propolis and in highlighting some critical aspects of research in the sector, among which in primis the absence of true clinical trials (with randomized controls) on humans. 

Author Response

(The authors gave the same response as above.)
